# Membrane procoagulation and N-terminomics/ TAILS profiling in Montreal platelet syndrome kindred with *VWF* p.V1316M mutation

Ejaife O. Agbani [1,2 ✉], Daniel Young[3], Si An Chen[3], Sophie Smith[3], Adrienne Lee [4], Alastair W. Poole[5], Antoine Dufour [1,3,9 ✉] & Man-Chiu Poon [6,7,8,9 ✉]

## Abstract

**Background** The Montreal platelet syndrome kindred (MPS) with VWF p.V1316M mutation (2B-VWDMPS) is an extremely rare disorder. It has been associated with macro-thrombocytopenia, spontaneous platelet clumping, mucocutaneous, and other bleeding, which can be largely prevented by von Willebrand factor (VWF) concentrate infusion. However, supplemental platelet transfusion has been required on occasion, particularly for severe gastrointestinal bleeds. This raised the question of whether a previously uncharacterized platelet dysfunction contributes to bleeding diathesis in 2B-VWDMPS patients. We have previously shown that membrane ballooning, a principal part of the platelet procoagulant membrane dynamics (PMD) after collagen stimulation, is driven by the influx of $Na^+$ and $Cl^-$, followed by the entry of water. **Methods** We study two members (mother and daughter) of the MPS kindred with severe bleeding phenotype and address this question by coupling quantitative platelet shotgun proteomics and validating biochemical assays, with the systematic analysis of platelet procoagulant membrane dynamics (PMD). Using N-terminomics/TAILS (terminal amine isotopic labeling of substrates), we compare changes in proteolysis between healthy and 2B-VWDMPS platelets. **Results** Here, we report in 2B-VWDMPS platelets, the loss of the transmembrane chloride channel-1 (CLIC1), and reduced chloride ion influx after collagen stimulation. This was associated with diminished membrane ballooning, phosphatidylserine externalization, and membrane thrombin formation, as well as a distinct phenotypic composition of platelets over fibrillar collagen. We also identify processing differences of VWF, fibronectin (FN1), and Crk-like protein (CRKL). 2B-VWDMPS platelets are shown to be basally activated, partially degranulated, and have marked loss of regulatory, cytoskeletal, and contractile proteins. **Conclusions** This may account for structural disorganization, giant platelet formation, and a weakened hemostatic response.

## Plain language summary

The Montreal platelet syndrome (MPS) is a very rare genetic illness caused by a specific modification in a protein called von Willebrand factor (VWF). VWF circulates in the blood and works with platelets to stop blood from escaping when blood vessels are injured. People with MPS have a bleeding problem, as they have decreased circulating VWF activity and platelets that also don't function as expected. Here, we studied a mother and a daughter who live with this condition to better understand if there are other reasons behind the bleeding issues in this family. These participants had low levels of several other proteins, and their platelets did not gather as usual to arrest bleeding. They also did not undergo the usual changes in shape. These changes could contribute to the bleeding problems reported in this family.

[1] Department of Physiology & Pharmacology, Cumming School of Medicine, University of Calgary, Calgary, AB, Canada. [2] Libin Cardiovascular Institute, University of Calgary, Calgary, AB, Canada. [3] McCaig Institute for Bone and Joint Health, University of Calgary, Calgary, AB, Canada. [4] Division of Hematology, Department of Medicine/Medical Oncology, University of British Columbia, Island Health, Victoria, BC, Canada. [5] School of Physiology, Pharmacology and Neuroscience, University of Bristol, Bristol, England, UK. [6] Division of Hematology & Hematological Malignancies, Department of Medicine, Cumming School of Medicine, University of Calgary, Calgary, AB, Canada. [7] Departments of Pediatrics and Oncology, Cumming School of Medicine, University of Calgary, Calgary, AB, Canada. [8] Arnie Charbonneau Cancer Institute, University of Calgary, Calgary, AB, Canada. [9] These authors jointly supervised this work: Antoine Dufour, Man-Chiu Poon. ✉email: ejaife.agbani@ucalgary.ca; antoine.dufour@ucalgary.ca; mcpoon@ucalgary.ca

While in the circulation, platelets are positioned proximal to the vessel wall which they continuously scan for injury. Blood vessel injury exposes sub-endothelial collagen, a potent and most abundant procoagulant agonist, which stimulates platelets to undergo dramatic procoagulant morphological transformations to support thrombin generation, recruit more platelets to the injury site, form fibrin mesh, and contract the first responding cells into a hemostatic plug that seals the wound and prevents further bleeding[1–4]. Emerging evidence shows that the human platelet assumes two major phenotypes with distinct functions during the hemostatic response: a population with elevated $Ca^{2+}$ and exposed phosphatidylserine (PS), which control thrombin and fibrin generation; and a second population with activated $\alpha_{IIb}\beta_3$ integrin which regulates aggregation and clot retraction[2,5]. The former population is exemplified by platelets with ballooned membrane upon collagen stimulation and may specifically assume a balanced non-spread (BNS) or ballooned procoagulant-spread (BAPS) phenotype on 3D collagen matrices. The latter population, however, are of a phenotype that is predominantly contractile, non-PS exposing, conventional-spread non-ballooned platelets (CSNB), and may assume typical 'fried-egg' shapes[2]. In this report, we reveal the attenuated formation of ballooned platelets coupled with increased formation of a distinct procoagulant CSNB platelet phenotype in the hemostatic response of platelets of members of the original Montreal platelet syndrome (MPS) kindred with von Willebrand factor (*VWF*) p.V1316M mutation[6,7]. This clinical phenotype, here abbreviated as 2B-VWD[MPS] or VWD[MPS], has been associated with macrothrombocytopenia, spontaneous platelet clumping in vitro, and severe mucocutaneous bleeding.

High-molecular-weight multimers (HMWM) of VWF which are 5000–10,000 kDa, are haemostatically the most active form of VWF[8]. In 2B-VWD HMWM VWF spontaneously bind to platelets and are cleared from circulation. The pathological increase in VWF-platelet binding leads to accelerated, proteolytic degradation of the large functional VWF mutimers by the plasma metalloproteinase ADAMTS13 to smaller, less adhesive, less haemostatically active multimers[8]. The deficiency of HMWM VWF therefore can result in bleeding complications ranging from epistaxis to gastrointestinal hemorrhage[8]. The 2B-VWD[MPS] patients examined in this study have a history of recurrent mucocutaneous bleeding, including angiodysplasia associated with gastrointestinal (GI) bleeding, in addition to bleeding from surgery/trauma/childbirth. The bleeding diathesis of this kindred was first reported in 1963[6,7,9]. For many years an intrinsic platelet defect was considered and was called MPS beginning 1979[6,7,10,11]. However, treatment of bleeding with platelet transfusion has not always been effective. Since the presence of 2B-VWD was clarified[6], VWF concentrate infusion has generally been more effective for the treatment and prevention of bleeding. However, supplemental platelet transfusion has been required on occasions, particularly for severe GI bleeds. This raises the question of whether a previously uncharacterized platelet dysfunction (in addition to macrothrombocytopenia) contributes to the bleeding diathesis in 2B-VWD[MPS] patients.

Here, we report on our findings in the 2 members of the MPS kindred with severe bleeding phenotype. We analyze protein expression and proteolysis (N-terminomics) by quantitative mass spectrometry, to determine changes in the platelet proteome and N-terminome in addition to demonstrating loss of regulatory, cytoskeletal, and contractile proteins that may account for the enlargement and structural disorganization in 2B-VWD platelets[12]. We also reveal a distinct phenotypic composition of 2B-VWD[MPS] platelets over collagen, and a weakened procoagulant remodeling, which we attribute to the loss of the transmembrane chloride channel CLIC1 and dysregulated proteolysis of VWF.

## Methods

Written informed consent was obtained in accordance with the Declaration of Helsinki. Blood samples were obtained from participants under the University of Calgary, Research Ethics Board approval (REB15-0550). All methods were performed in accordance with the Alberta Health Services and The University of Calgary research guidelines and regulations.

**The 2B-VWD[MPS] patients and healthy controls**. Eight previous studies have reported on the two patients of the present study;[6,7,10,11,13–16] (Supplementary Data 1). The two 2B-VWD[MPS] patients in this study are patients L.T.B. (current age 81 years) and her daughter I.B. (age 64) reported previously[6,11]. Both were in the original report in 1963[7] and both participated in Frojmovic's original MPS studies beginning in 1979[10]. Also, both 2B-VWD[MPS] patients were on Atorvastatin, a lipid-modifying drug and an anti-angiogenesis agent for GI angiodysplasia. In addition, the mother has hypertension, which was controlled with a calcium channel blocker Amlodipine, and the daughter is non-hypertensive. Importantly, neither of the 2B-VWD[MPS] patients studied has other comorbidity such as diabetes, renal, or rheumatologic problems. Bleeding incidence and phenotype was similar in L.T.B and I.B. although L.T.B alone was on twice weekly prophylactic VWF concentrate infusions. Blood samples were obtained during their non-bleeding state and the L.T.B.'s samples of this study were obtained 4 days after the previous VWF infusion. Four healthy participant controls (HPC) were females aged 72.3 ± 7.1(SD).

**Platelet-rich plasma (PRP) preparation**. Blood was collected into blue-capped sodium citrate tubes (0.109 M / 3.2%) and centrifuged at 180 x $g$ for 17 min to prepare platelet-rich plasma (PRP).

**Washed platelets and lysate preparation for protein assays**. Platelet-rich plasma was centrifuged at 650 x $g$ for 10 min in the presence of 10 µmol/L indomethacin and 0.02 U/mL apyrase[2]. Platelets were lysed with buffer containing 1% SDS, 200 mM HEPES (pH 8.0), 100 mM ammonium bicarbonate, 10 mM EDTA, and protease inhibitor cOmplete™ tablets (Roche, Mississauga, ON, Canada). Protein precipitation was performed with 600 µL of ice-cold acetone, incubated at −20 °C overnight, and followed with centrifugation at 8000 x $g$ for 10 min.

**Confocal microscopy**. Confocal images were acquired using a Nikon A1R laser scanning confocal microscope. The system is equipped with 405 nm, 488 nm, 561 nm, and 640 nm solid-state lasers, a fully motorized inverted frame, and high-precision stage for multi-focus and multi-point imaging, piezo drive, and Nikon Perfect Focus System (PFS) for drift-free imaging. Images were captured at Nyquist by Nikon NIS-Elements imaging software, and by means of an oil immersion Plan Apo Lambda objective lens (60x; Numerical Aperture: 1.4; Working Distance: 0.13 mm). The acquisition involved fast/sensitive 4-color and transmitted light. The acquisition setting was kept constant at high-speed/high-definition resonant scanning (up to 1024 × 1024 pixels). In some experiments (4D), z-stack images (xyz, 3D) were acquired over time (t).

**Fluorescence imaging of platelets**. We visualize in plasma, platelets of the two 2B-VWD[MPS] patients and four healthy participant controls (HPC) adherent to surfaces precoated with inert serum albumin (BSA), fibrillar collagen, or human fibrinogen. The intensity of membrane P-selectin and PS expression were monitored through the binding of Alexa-Fluor® 488 Anti-human

CD62P antibody, and Alexa-Fluor® 568 conjugated annexin-V, respectively. To detect thrombin generation on the platelet membrane, we used Alexa-Fluor® conjugated mouse monoclonal antibody specific for an epitope mapped between amino acids 331–376 within an internal region of human thrombin. We then performed a systematic analysis of platelet procoagulant membrane dynamics (PMD), platelet quantitative shotgun proteomics, and biochemical studies[2,3,17].

**Terminal amine labeling of substrates and shotgun proteomics.** Platelet lysates from our 2B-VWD[MPS] patients and healthy participant controls (HPC) were prepared in SDS lysis buffer (1% SDS, 100 mM Ammonium bicarbonate, 1 mM EDTA, 1x protease inhibitors [Roche cOmplete, mini EDTA free protease inhibitor cocktails]). The lysates were sonicated with a probe-type sonicator at intensity 5 for 5 s twice to clarify our lysate; after which, the samples were centrifuged at 14,000 x g for 10 min and the supernatants were transferred to a new Protein LoBind tube before a bicinchoninic acid (BCA) protein assay was conducted according to the manufacturer's directions (Thermo Fisher catalog 23227). 500 μg of total protein per condition was removed for TAILS analysis. The samples were topped up to 1 mL with SDS lysis buffer. The samples were then treated with 5 mM of dithiothreitol (DTT) and incubated at 37 °C for 1 h to reduce disulfide bonds. Free thiols on cysteines were blocked by adding 15 mM of iodoacetamide for 25 min in the dark at room temperature. The thiol-blocking reaction was then quenched by adding an additional 5 mM of DTT. The samples were labeled with 40 mM of light formaldehyde for the controls, or heavy formaldehyde for 2B-VWD[MPS] (Cambridge Isotope Laboratories, Inc. catalog DLM-805-PK), immediately followed by the addition of 20 mM of sodium cyanoborohydride. The pH was adjusted to 6.5 before incubation overnight at 37 °C. The next day, an additional 10 mM of light or heavy formaldehyde was added followed by 10 mM of sodium cyanoborohydride. Then, the samples were pH balanced to 6.5. Samples were incubated at 37 °C for 1 h. After incubation, control, and experimental pairs were combined in a 50 mL Falcon tube with 40 mL of 8:1 ice-cold acetone methanol and incubated at −20 °C for 4 h. The samples were washed in 100% methanol three times by centrifuging at 9000 x g for 15 min. Following the last wash, the supernatant was discarded, and the samples were air-dried in a fume hood for 5 min to remove residual methanol. The samples were then suspended in 200 μL of 200 μM NaOH and transferred to a 1.5 mL protein LoBind tube before vortexing for 15 min. After the protein pellet was completely resuspended, 300 μL of 200 mM HEPES buffer was added, followed by the addition of 25 μg of mass spec grade trypsin (Thermo Fisher, catalog no 90058). The pH was adjusted to ~8 before incubation overnight at 37 °C. 10% of each sample was removed for preTAILS (shotgun) analysis. The preTAILS samples were acidified to a pH of less than 3 with trifluoracetic acid (TFA) and stored at 4 °C overnight for SEP-PAK cleanup alongside the TAILS samples. The remaining 90% of each sample was kept for TAILS analysis. 15 μg of TAILS HPG-ALD polymer (https://ubc.flintbox.com/technologies/888fc51c-36c0-40dc-a5c9-0f176ba68293) was added followed by 40 mM of sodium cyanoborohydride. The pH was adjusted to pH ~6.5 before incubation at 37 °C overnight. 400 μL of 200 mM NaOH was added to a 10,000 Nominal Molecular Weight Limit (NMVL) Amicon filtration tube and centrifuged for 5 min at 5000 x g. The NaOH wash step was repeated twice for a total of three washes. The filtration tube was then washed with 500 μL of HPLC water 3 times using the same centrifuge settings. The TAILS samples were loaded into the Amicon filtration tube and centrifuged at 10,000 x g, eluate was collected in a new tube. 500 μL of the sample was filtered at a time

until the full volume of each sample had been filtered. Finally, 200 μL of 1 M Tris-HCl (pH 6.8) was used to wash residual peptides from the filtration column. Samples were then acidified to a pH less than 3 with TFA. PreTAILS and TAILS samples proceeded to SEP-PAK (C18) cleanup (Waters catalog no. WAT020515). 3 mL of conditioning solution (90% v/v methanol, 0.1% TFA v/v, 10% v/v HPLC water) was passed through the solid phase extraction (SPE) column, followed by 2 mL of load solution (100% HPLC water, 0.1% v/v TFA). The sample was loaded into the SPE column, followed by an additional 1 mL of load solution. Salt contaminants were removed by passing 3 mL of desalt solution (5% v/v. methanol, 95% v/v HPLC water, 0.15 TFA) through the SPE column. Samples were collected in a new protein LoBind tube by passing 1 mL of elution solution (50% v/v HPLC water, 50% v/v acetonitrile, 0.1% TFA) through the SPE. Three holes were poked into the lid of each protein LoBind tube, and the samples were snap-frozen in liquid nitrogen prior to being dehydrated in the SpeedVac over the period of 8 h. Finally, samples were frozen at −80 °C before submission for liquid chromatography-tandem mass spectrometry (LC-MS/MS).

**High-performance liquid chromatography (HPLC) and mass spectrometry.** All liquid chromatography and mass spectrometry experiments were carried out by the Southern Alberta Mass Spectrometry core facility at the University of Calgary, Canada. Analysis was performed on an Orbitrap Fusion Lumos Tribrid mass spectrometer (Thermo Fisher Scientific, Mississauga, ON) operated with Xcalibur (version 4.0.21.10) and coupled to a Thermo Scientific Easy-nLC (nanoflow Liquid Chromatography) 1200 system. TAILS or Tryptic peptides (2 μg) were loaded onto a C18 trap (75 μm × 2 cm; Acclaim PepMap 100, P/N 164946; Thermo Fisher Scientific) at a flow rate of 2 μL/min of solvent A (0.1% formic acid and 3% acetonitrile in LC-mass spectrometry grade water). Peptides were eluted using a 120 min gradient from 5 to 40% (5% to 28% in 105 min followed by an increase to 40% B in 15 min) of solvent B (0.1% formic acid in 80% LC-mass spectrometry grade acetonitrile) at a flow rate of 0.3% μL/min and separated on a C18 analytical column (75 μm × 50 cm; PepMap RSLC C18; P/N ES803; Thermo Fisher Scientific). Peptides were then electrospray using 2.3 kV into the ion transfer tube (300 °C) of the Orbitrap Lumos operating in positive mode. The Orbitrap first performed a full mass spectrometry scan at a resolution of 120,000 FWHM to detect the precursor ion having a mass-to-charge ratio (m/z) between 375 and 1575 and a +2 to +4 charge. The Orbitrap AGC (Auto Gain Control) and the maximum injection time were set at $4 \times 10^5$ and 50 ms, respectively. The Orbitrap was operated using the top speed mode with a 3 s cycle time for precursor selection. The most intense precursor ions presenting a peptidic isotopic profile and having an intensity threshold of at least $2 \times 10^4$ were isolated using the quadrupole (isolation window of m/z 0.7) and fragmented with HCD (38% collision energy) in the ion routing Multipole. The fragment ions (MS2) were analyzed in the Orbitrap at a resolution of 15,000. The AGC, the maximum injection time, and the first mass were set at $1 \times 10^5$, 105 ms, and 100 ms, respectively. Dynamic exclusion was enabled for 45 s to avoid the acquisition of the same precursor ion having a similar m/z (±10 ppm).

**Proteomic Data and Bioinformatic Analysis.** The first comprehensive and quantitative analysis of human platelet protein composition demonstrated a comparative analysis of structural and functional pathways[18]. Here, spectral data were matched to peptide sequences in the human UniProt protein database using the MaxQuant software package v.1.6.0.1, peptide-spectrum match false discovery rate (FDR) of <0.01 for the shotgun

**Fig. 1 Platelet clumping in 2B-VWD of the original Montreal platelet syndrome (2B-VWD^mps) kindred with *VWF* p.v1316m mutation.** Citrated platelet-rich plasma (PRP) from two 2B-VWD^MPS patients and four healthy participant controls (HPC) were left to adhere to inert bovine serum albumin (BSA; **a–d**), fibrinogen (**e**). In **a**, **b** platelets were labeled with Alexa-fluor 488 anti-human CD62P (P-Selectin), Alexa-fluor 568 Annexin-V and Alexa-fluor 405-conjugated CD42b-human GPIb alpha antibody. **c** plot of mean fluorescent signal intensities of P-selectin, thrombin, and PS markers in HPC and 2B-VWD^MPS platelets adherent on BSA-coated surfaces. **d** Plot of platelet size. The chart in **e** is the same as in **c**, but for platelets adherent on fibrinogen-coated surfaces. Data (**c–e**) are presented as box-and-whiskers plots. Images (**a**, **b**) were captured at Nyquist. Scale bars: 7 μm (**a**, **b**).

proteomics data and <0.05 for the N-terminomics/TAILS data. Search parameters included a mass tolerance of 20 p.p.m. for the parent ion, 0.05 Da for the fragment ion, carbamidomethylation of cysteine residues (+57.021464), variable N-terminal modification by acetylation (+42.010565 Da), and variable methionine oxidation (+15.994915 Da). For the N-terminomics/TAILS data, the cleavage site specificity was set to semi-ArgC (search for free N-terminus) for the TAILS data and was set to ArgC for the preTAILS data, with up to two missed cleavages allowed. Significant outlier cut-off values were determined after log(2) transformation by boxplot-and-whiskers analysis using the BoxPlotR[19,20]. Database searches were limited to a maximal length of 40 residues per peptide. Peptide sequences matching reverse or contaminant entries were removed.

**Reactome pathway analysis.** To identify interconnectivity among statistically changing proteins, the STRING-db (Search Tool for the Retrieval of Interacting Genes) database was used to identify interconnectivity among proteins. The protein-protein interactions are encoded into networks in the STRING.v11 database (https://string-db.org)[21,22]. Metascape[23] (https://metascape.org) analysis was used to identify changes in functional enrichment, interactome analysis, and gene annotation. Our data were analyzed using *Homo sapiens* as our model organism at a false discovery rate of 1%.

**Statistical analysis.** For the proteomics and N-terminomics analysis, an interquartile boxplot analysis was applied to determine the differential enrichment/upregulation of proteins or peptides[19]. The FDR was generated by the bioinformatic tools used for pathway analysis.

**Reporting summary.** Further information on research design is available in the Nature Portfolio Reporting Summary linked to this article.

## Results and discussion

**Platelets characterization of 2B-VWD^MPS patients.** Outcomes of PMD analysis are comparable between both 2B-VWD^MPS patients of this study. Freshly isolated 2B-VWD^MPS platelets are basally already activated, significantly larger, and exhibited platelet clumps when compared to platelets of the healthy participant controls (HPC) (Fig. 1a–c). Both patients had persistent platelet clumping irrespective of bleeding status. The image in Fig. 1b shows individual platelets and platelet clumps in PRP of 2B-VWD^MPS over serum albumin substrate, indicating the

clumps were present at the time that PRP was prepared (Mean ± SD, HPC: 1.6 ± 1.2 vs. 2B-VWD^MPS: 20.4 ± 3.7). The fluorescence signal quantified in Fig. 1c is derived from both, individual platelets and platelet clumps as shown in Fig. 1b. Also, we occasionally sighted heterotypical aggregates of platelets and leukocytes in 2B-VWD^MPS but not HPC PRP (data not shown).

Platelet counts are for the most part impractical to enumerate because of clumping; however, when enumerable, counts remain stable but low during the study period, irrespective of bleeding status (Mean ± SD: L.T.B: 25 ± 2, I.B: 29 ± 3 × 10⁹/L). We did not repeat multimer studies during the bleeding state as a matter of routine; we supposed that multimer studies are qualitative showing loss of higher multimers but are not quantitative enough to evaluate clinical changes.

Under basal (unstimulated, Fig. 1a–d) conditions, the membranes of 2B-VWD^MPS platelets expressed low, but still significantly higher levels of P-selectin, and higher bound annexin-V and VWF, compared to HPC platelets. (Fig. 1a–c). These findings agree with the report of Casari et al.[24], that shows increased activation in suspended platelets from other 2B-VWD patients with p.V1316M. With fibrinogen stimulation Fig. 1e, 2B-VWD^MPS platelets show incremental activation, but lesser membrane procoagulation (PS exposure + thrombin formation) when compared to HPC platelets (Fig. 1e). Attenuated activation after fibrinogen stimulation is likely indicative of decreased integrin $\alpha_{IIb}\beta_3$ activation as previously reported in platelets from MPS patients[25]. The basal preactivated states of 2B-VWD^MPS platelets, coupled with an incremental response to normal fibrinogen from human plasma, may account for the propensity to micro-aggregate and clump formation (Fig. 1b). Additionally, calpain-1 and the calpain small subunit 1 (CAPNS1) required for calpain activity are downregulated in 2B-VWD^MPS platelets as shown by both our shotgun proteomics preTAILS and TAILS data (Supplementary Data 2–5). This supports the previous report of abnormally low levels of calpain-1 and up to 70% decrease in calpain activity in 'MPS platelets'[11]. Loss of the proteolytic activity of calpain may also contribute to reduced procoagulant remodeling[11].

Interestingly our proteomic analysis also shows that compared to controls, platelet α-granules of 2B-VWD^MPS patients contain nearly twice as much prothrombin, the inactive precursor of thrombin[26], still plasma levels show a fold decrease in prothrombin, and prothrombin activation peptide fragment 1 + 2, thrombin light chain and thrombin heavy chain. This may be indicative of some dysfunction in granular secretion and is consistent with defects in procoagulant activity observed in

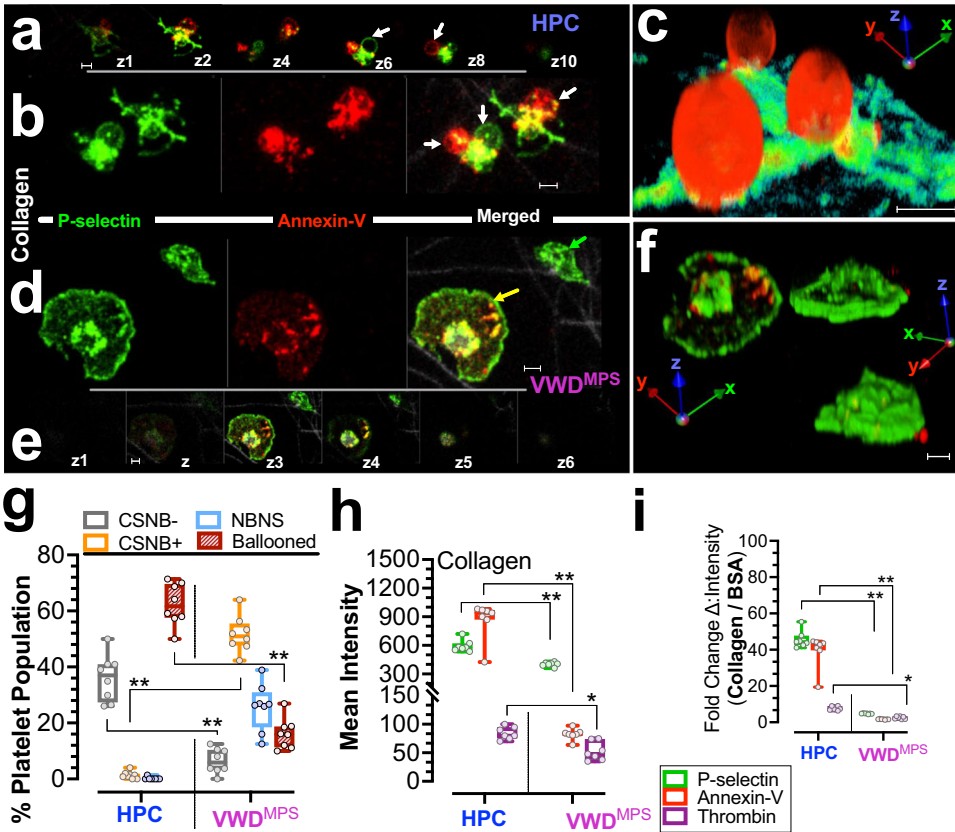

**Fig. 2 Attenuated collagen-induced membrane ballooning in platelet derived from 2B-VWD of the original Montreal platelet syndrome (2B-VWD^mps)** **kindred with *VWF* p.v1316m mutation.** Citrated platelet-rich plasma (PRP) from two 2B-VWD^MPS patients and four healthy participant controls (HPC) were allowed to adhere to fibrillar collagen. Extended focus images at the 45-min timepoint are shown. In **a–f**, platelets were labeled with Alexa-fluor 488 anti-human CD62P (P-Selectin), Alexa-fluor 568 Annexin-V and Alexa-fluor 647 conjugated thrombin antibody. **a** shows z-sections separated by 2 µm, of the ballooned HPC platelet shown in **c**. In **a**, **b** the white arrow points to ballooned platelets. **c** representative 3D reconstruction of ballooned platelets adherent over fibrillar collagen. **d** shows the conventional-spread non-ballooned platelets (CSNB), one annexin positive (CSNB+, yellow arrow), and one annexin negative (CSNB-, green arrow). **e** shows z-sections separated by 1 µm, of the CSNB 2B-VWD^MPS platelet shown in **d**. **f** show 3D reconstruction of CSNB platelets adherent over fibrillar collagen. **g** percent plot of phenotypes in HPC and 2B-VWD^MPS platelets. **h** plot of mean fluorescent signal intensities of P-selectin, thrombin, and PS markers in HPC and 2B-VWD^MPS platelets adherent on collagen-coated surfaces. **i** show fold-change in signal intensities derived by dividing the intensity after collagen stimulation by mean intensity under BSA (Fig. 1). Data are presented as box-and-whiskers plots. Images were captured at Nyquist. Scale bars: 4 µm (**a–f**).

2B-VWD^MPS platelets. Studies examining differential granular secretion in 2B-VWD^MPS platelets will elucidate this.

The dramatic inflation of platelet membranes into ballooned structures (ballooning) has been previously reported[27,28], characterized[2,29], and shown to increase the membrane evaginated for PS exposure[2–4,17,29]. Thus, ballooning increases the catalytic surface area for the docking of activated coagulation factors V and X, and the formation of prothrombinase complex on the platelet membrane[2,3,30–34], and serves to amplify the coagulation process[31,32,34]. Up to 60% of healthy human platelets undergo membrane ballooning once adherent to fibrillar collagen (Fig. 2a–c), and about 40% remain Conventionally Spread Non-Ballooned (CSNB)[2,29]. Moreover, platelet heterogeneity in collagen-dependent in vitro and in vivo thrombus formation in flowing blood remains intriguing[35].

CSNB platelets are typically annexin-V negative, contractile, and non procoagulant[2,5] after collagen stimulation. In this study, however, after collagen stimulation, 2B-VWD^MPS platelets are shown to be majorly procoagulant CSNB (with PS exposure; CSNB+; yellow arrow; Fig. 2d, compared to CSNB-, green arrow). Upon exposure to fibrillar collagen, while only 16.2 ± 5.5 (%, SD) of 2B-VWD^MPS platelets ballooned

compared to 62.4 ± 7.5 (%, SD) in HPC platelets (compare Fig. 2a–c with d–f, then see Fig. 1) as much as 51.7 ± 6.5 (%, SD) were CSNB+ as compared to only 1.3 ± 1.3 (%, SD) in HPC platelets (compare Fig. 2a–c with d–f, then see Fig. 2g). This phenotypic shift in 2B-VWD^MPS platelets composition over collagen, from predominantly 'ballooned' to 'spread' procoagulant platelets may represent an alternate albeit inefficient hemostatic response of platelets lacking the protein repertoire for full procoagulant response (Fig. 3a–c, Supplementary Data 2–6).

In the 2B-VWD^MPS patients, it may contribute to a reduced hemostatic response after vessel injury[2,29]. Additionally, 2B-VWD^MPS platelets show a further increase in P-selectin and PS expression, and membrane thrombin formation after collagen stimulation (Fig. 2h, compared to basal Fig. 1c). However, both the sum intensities of the fluorescent signals of these proteins (Fig. 2h), and the fold-change or absolute response to collagen stimulation are significantly attenuated (Δ; Fig. 2i). Expectedly[24], 2B-VWD^MPS platelets form smaller aggregates over fibrillar collagen when compared to HPC platelets after adjustment for platelet number (Fig. 3a–c, also see (Fig. 1d, also see Supplementary Movie 1).

**Fig. 3 Diminished aggregation response to collagen in platelet derived from 2B-VWD of the original Montreal platelet syndrome (2B-VWD^mps) kindred with *VWF* p.v1316m mutation.** Citrated platelet-rich plasma (PRP) from two 2B-VWD^MPS patients and four healthy participant controls (HPC) were allowed to adhere and aggregate over collagen-collated surfaces. In **a** and **c**, platelets were labeled with Alexa-fluor 488 anti-human CD62P (P-Selectin) and Alexa-fluor 568 Annexin-v. **a** and **c** show 3D rendition of HPC (**a**) and 2B-VWD^MPS (**c**) platelet aggregates. Chart **b** is a plot of aggregate volumes on the coated surface. Data (**b**) are presented as box-and-whiskers plots. Images were captured at Nyquist. Scale bars: 4 μm (**a** and **c**).

**N-terminomics and proteomics analysis of platelets**. Combined quantification of the global proteome, phosphoproteome, and proteolytic cleavage have been used to characterize platelet dysfunction in the human bleeding disorder associated with mutations in the gene encoding anoctamin-6[36]. As proteases play key biological roles in coagulation, we characterize the N-terminome and proteolytic profile, as previously done[37,38] (Supplementary Data 2–4). To investigate if 2B-VWD^MPS have proteolytic defects and potentially identify their distinct substrates, we subject their platelets and healthy platelets to an N-terminomics/TAILS and pre-enrichment TAILS/shotgun proteomics analysis (Fig. 4a and Supplementary Figs 1–2). Platelet protein lysates (n = 4) are denatured, alkylated, and the peptide α- and ε-amines are isotopically labeled with heavy or light formaldehyde[39,40]. Following trypsin digestion, negative selection against unlabeled α-amines of trypsinized peptides is attained by incubating the samples with a 3-fold excess of the dendritic polyglycerol aldehyde TAILS polymer[39] (Fig. 4a). In our pre-enrichment TAILS shotgun proteomics data, we identify enrichment of inflammatory responses in 2B-VWD^MPS platelets, in addition to reduced granule contents (Fig. 4b) and defects in cell-substrate adhesion using metascape[23] (Supplementary Figs. 1–2 and Supplementary Data 2–5).

The stunted response of 2B-VWD^MPS platelets to collagen stimulation is consistent with a partially degranulated or 'exhausted' state of preactivated platelets, as we confirm through the quantitative shotgun proteomic analysis of α-granule markers of platelet activation (Fig. 4b). We next perform additional pathway enrichment analyses using another tool, STRING-db[21], and we also identify enrichment of cell-extra-cellular matrix (ECM) interactions. These data suggest that diminished platelet activation and adherens junction enrichment pathways are detected in 2B-VWD^MPS platelets (Supplementary Fig. 1b, c). Over 667 unique N-termini are identified using TAILS (Fig. 4c and Supplementary Data 2–5).

Interestingly, platelets of 2B-VWD^MPS patients show a distinct N-terminome and proteolytic profile as compared to HPC platelets (Fig. 4d and Supplementary Fig. 2), suggestive of dysregulated platelet proteolysis as an additional dysfunction in 2B-VWD^MPS. Our N-terminomics/TAILS analysis identifies several statistically changing cleaved proteins using boxplot interquartile analysis. These include von Willebrand Factor (VWF) at $^{1780}R{\downarrow}Y^{1781}$, fibronectin (FN1) at $^{1365}R{\downarrow}F1^{366}$, and Crk-like protein (CRKL) at $^{290}H{\downarrow}V^{291}$, only in 2B-VWD^MPS platelets (Fig. 4e and Supplementary Data 5–6). Firstly, cleavage of VWF at $^{1780}R{\downarrow}Y^{1781}$ could result in the production of lower-molecular-weight multimers (LMWM) of VWF, with less hemostatic capacity[8]. Additionally, this cleavage of VWF corresponds to the mid-point of the collagen-binding domain on VWF which could lead to defects in platelet activation.

Indeed, relative to HPC platelets, platelets of 2B-VWD^MPS show marked losses of regulatory, cytoskeletal, and structural proteins integral to the procoagulant morphological transformation of platelets (Supplementary Fig. 1a–c)[2,3]. For example, we note >2.9 mean $Log_2$ fold decrease in actinin and actin proteins (Fig. 4f, g), which we validated (Fig. 4h). These losses and the associated attenuation of membrane ballooning and procoagulation in 2B-VWD^MPS platelets, mimick outcomes of experiments in which we use jasplakinolide to alter actin function and thereby inhibit membrane ballooning[2].

The structural protein loss[41] in 2B-VWD^MPS (Fig. 4f–h), may account for the increased size of 2B-VWD^MPS platelets (Fig. 1d) and the structure disorganization previously reported[41,42]. Rand et al.[42], show that the platelet size in median forward scatter (flow cytometry) of 3 members (including the two studied here) of this kindred at 39,900 ± 5200 (mean ± SEM) was much larger than that of the other six 2B-VWD (11,100 ± 2200), including 3 with *VWF* p.V1316M. Here, we also observe a > 3.5 mean $Log_2$ fold decrease in myosin light chain and tubulin proteins (Fig. 4f–h). The loss of these contractile and regulatory machinery proteins[43] (Supplementary Data 6) likely contributes to the diminished capacity of 2B-VWD^MPS platelets to remodel or undergo procoagulant ballooning after collagen stimulation. Also, knocking-out platelet tubulin genes has previously been shown to result in platelet dysfunction and increased bleeding times[44].

We next wanted to know; what mechanisms underlie the stunted procoagulant response of 2B-VWD^MPS platelets to collagen? We previously revealed the molecular mediators of the events regulating the transformation of human platelets into a ballooned phenotype (BAPS and BNS) during the hemostatic response to exposed sub-endothelial collagen[2,3]. Here we show that ballooning is predominantly induced by the collagen receptor glycoprotein VI (GPVI) activation and driven by a coordinated influx of $Na^+$ and $Cl^-$, followed by the entry of osmotically obliged water through water channel aquaporin-1[2,45].

Our quantitative proteomics analysis of 2B-VWD^MPS platelets reveals up to a >4 mean $Log_2$ fold decrease in chloride intracellular channel protein 1 (CLIC1; Fig. 4f, g). CLIC1 has been shown to localize to the platelet membrane[46], with mRNA levels of CLIC1 up to 37-fold aquaporin-1 levels in human platelets[47]. The conditional CLIC1 knock-out mouse model was shown to result in mild platelet dysfunction characterized by prolonged bleeding times and decreased platelet activation[46].

Using western blotting, we validate the decreased expression of CLIC1 in 2B-VWD^MPS platelets (Fig. 4h and Supplementary Fig. 3) and demonstrate reduced $Cl^-$ influx into 2B-VWD^MPS platelets after collagen stimulation (Fig. 5a–d). We use N-(Ethoxycarbonylmethyl)-6-methoxyquinolinium bromide (MQAE) dye, which detects intracellular $Cl^-$ influx through diffusion-limited

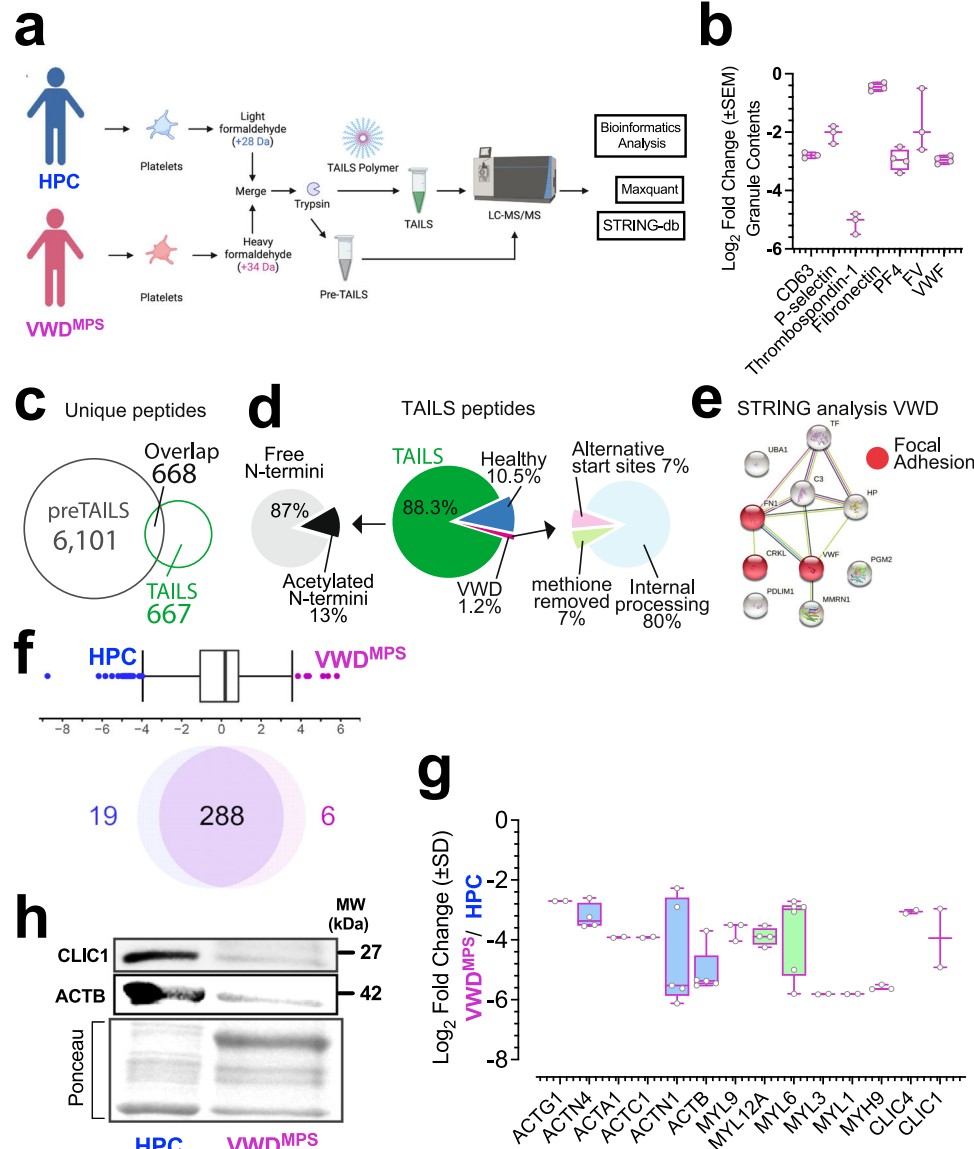

**Fig. 4 N-terminomics and validation data of platelet derived from 2B-VWD of the original Montreal platelet syndrome (2B-VWD^mps) kindred with VWF p.V1316m mutation. a** Experimental design of the N-terminomics/TAILS and shotgun/preTAILS workflow. Data in **b**–**h** are from washed platelet lysates of healthy controls and 2B-VWD^MPS patients. Isolated proteins were analyzed by N-terminomics/TAILS and shotgun/preTAILS analysis. Control samples were labeled with isotopically light formaldehyde and VWD samples were labeled with isotopically heavy formaldehyde. Peptide and protein identification was analyzed using Maxquant version1.6.0.1. **b** log2 fold-change (unstimulated) graph of platelet alpha and dense granule proteins identified in quantitative shotgun proteomics analysis. **c** The numbers of identified and shared peptides in each condition are shown. **d** Distribution of free N-termini, α-amine acetylated N-termini within the TAILS samples (*Left*), breakdown of identified peptides associated with each condition based on 0.5-2-Fold-change cut-off values, Green non-changing peptides, blue healthy, red 2B-VWD^MPS (*Center*). Distribution of N-termini and their modifications, and their cleavage patterns within the 2B-VWD^MPS sample (*Right*). **e** String-DB pathway analysis of identified substrates upregulated in 2B-VWD^MPS patients. **f** Venn diagram showing results of quantitative shotgun platelet proteomics analysis and group comparison as illustrated. Interquartile boxplot analysis was performed to identify as outliers, proteins enriched, and downregulated in 2B-VWD^MPS platelets. **g** log2 fold-change graph of select proteins downregulated in 2B-VWD^MPS platelets. **h** Western blot analysis of 2B-VWD^MPS and HPC platelets lysates validating a diminished expression of CLIC1, total actin, and Ponceau dye for protein loading. Data (**b**, **g**) are presented as box-and-whiskers plots.

collisional quenching[48], as a fluorescent indicator of intracellular Cl⁻ (Fig. 5a–d).

In Fig. 5c, suspended HPC but not 2B-VWD^MPS platelets show a significant loss of MQAE intensity due to Cl⁻ influx and increased intracellular chloride concentration $[Cl^-]_i$. Parallel immunofluorescence experiments show loss of MQAE dye in adherent, activated, and procoagulant platelets only (comparing Fig. 5a, b). Pre-incubation of HPC platelets with the chloride channel blockers 4,4′-diisothiocyano-2, 2′-disulfonic acid stilbene

(DIDS, 100 μM) or niflumic acid (NFA, 100 μM), reduce collagen-induced Cl⁻ influx, and stabilized MQAE intensity (Fig. 5c, d). These data indicate that loss of Cl⁻ channel in platelets of 2B-VWD^MPS limits the intracellular Cl⁻ entry required for a full procoagulant response in platelets[2,49].

Overall, our data does not suggest whether the changes in the MPS platelets we describe in this report occurred because of the 'clumping process' of circulating platelets or whether they arise during megakaryocyte maturation and/or platelet

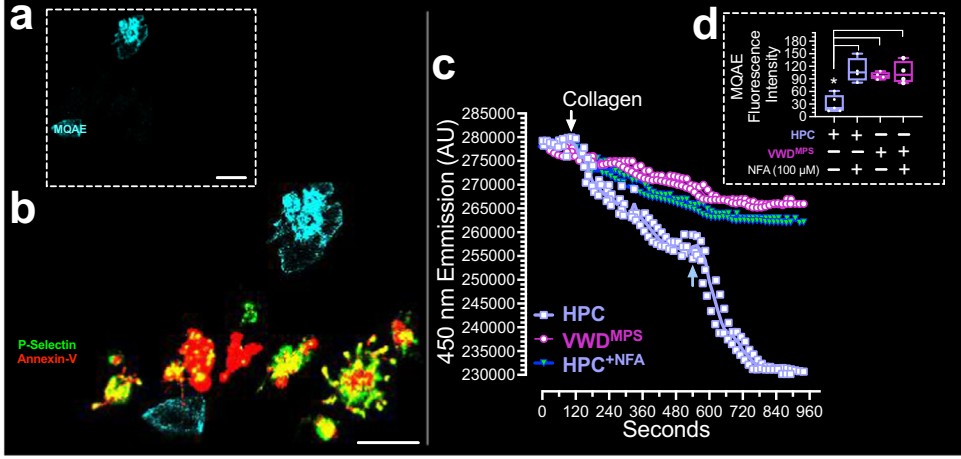

**Fig. 5 Reduced chloride ion influx after collagen stimulation in platelet derived from 2B-VWD of the original Montreal platelet syndrome (2B-VWD^mps) kindred with *VWF* p.V1316m Mutation.** Plasma platelets were preloaded with the chloride ion (Cl⁻) indicator, N-(Ethoxycarbonylmethyl)-6-methoxyquinolinium bromide (MQAE) (**a**, **b**), which detected Cl- entry by diffusion-limited collisional quenching. Thus, decreases in fluorescence intensity against time represented increases in intracellular Cl⁻ concentration over time, as recorded in suspended platelets (**c**, **d**), and after 30 min in immunofluorescence assay (**a**, **b**). **d** Plot of MQAE intensity 30 min after either treated or untreated plasma platelets were exposed to collagen. The blue arrow in **c** indicates the probable onset of membrane ballooning. Data are presented as box-and-whiskers plots. Images were captured at Nyquist. Scale bars: 4 µm (**a** and **b**).

biogenesis. We, however, speculate that these changes are more likely the consequence of the latter, and Nurdens et al. have reported data showing defective megakaryopoiesis in patients with 2B-VWD[12,50]. Further studies are now needed to clarify this.

## Conclusion
We confirm in 2B-VWD^MPS, a rare bleeding disorder, a phenotype of basally activated, partially degranulated, and microaggregating platelets with reduced membrane ballooning and procoagulant response to collagen. In addition, we show in 2B-VWD^MPS platelets, the loss of intracellular chloride channel CLIC1 and a diminished Cl⁻ influx after collagen stimulation. There is also a loss of cytoskeletal proteins integral to the procoagulant morphological transformation of platelets. We propose that these platelet dysfunctions and disorganization coupled with dysregulated protease activity contribute to a diminished hemostatic response and the severe bleeding diathesis seen in our 2B-VWD^MPS patients. Whether the findings in this MPS family with 2B-VWF due to *VWF* V1316M mutation are common to 2B-VWD with other *VWF* mutations remains to be studied.

## Data availability
All source data are available in Supplementary Data 7. The full proteomics data are publicly available. The data were deposited in PRIDE Archives, accession number: **PXD035713** (ProteomeXchange accession: PXD035713). Project Webpage: http://www.ebi.ac.uk/pride/archive/projects/PXD035713.

## Code availability
R codes and data are available from Antoine Dufour (antoine.dufour@ucalgary.ca) upon reasonable request.

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

## Acknowledgements

This work was supported by the Live-Cell Imaging Facility, funded by the Snyder Institute at the University of Calgary. EOA is supported by the Cumming School of Medicine, University of Calgary, Alberta, Canada. We thank Luiz Gustavo N. de Almeida and Isabella Orchard for their assistance. A.D. was supported by an NSERC Discovery Grant (DGECR-2019-00112), Alberta Innovates AICE concepts (202102341), and Canadian Institutes of Health Research (449589).

## Author contributions

Study conceptualization (M.P., A.L., E.O.A.); Patient recruitment (M.P., A.L.); Study/Experimental design and supervision (E.O.A., A.D.). Experimentation (E.O.A., S.C., D.Y., S.S.); Data analysis (D.Y., A.D., E.O.A.); Statistical analysis (E.O.A.), Discussion (E.O.A., A.D., A.L., A.W.P., & M.P.). Manuscript preparation (E.O.A.) and revision (E.O.A., A.D., A.L., A.W.P., M.P.).

## Competing interests

The authors declare no competing interests.
