## [Peer Review File · Communications Medicine]

Reviewers' comments:

Reviewer #1 (Remarks to the Author):

The authors have studied two members of a family with what was historically termed Montreal platelet syndrome (MPS) and then shown to be von Willebrand disease type 2B (Type 2B VWD) caused by a heterozygous VWFV1316M mutation. Man Chiu Poon has a history of publications on MPS and so knows the disorder well. Combining his experience with the groups of A Dufour (N-terminomics) and A. Poole (procoagulant activity) has enabled a unique look at two members of the original family with this rare platelet disorder and provided some stimulating results. The manuscript is very well written and provides a new insight on MPS. At the same time, the authors have been highly selective with regard to the results presented.

1. Two members of the original family designated as MPS, a mother and her daughter, featured in this study. In view of their age, it would be interesting to know if they were among the original patients studied by Frojmovic and Milton in their original reports on this disease. In fact it would be useful to have in the Supplemental data a list of all previous publications featuring these 2 patients. Enigmatically, the authors fail to cite a highly relevant study performed by Okita et al (Okita JR, Frojmovic MM, Kristopeit S, Wong T, Kunicki TJ. Montreal platelet syndrome: a defect in calcium-activated neutral protease (calpain). *Blood* 1989; 1:715-721). Were the current patients a part of the Okita study? Either way, this publication must be cited for calpain (a calcium-activated protease) activity may well account for the decreased amount of selected cytoskeletal and other proteins in the platelets of the patients as described in the current report. Was calpain normally detected in the proteomic studies? Do they agree with Okita et al that it is decreased in the platelets of the patients studied? Possibly it becomes used up as the MPS platelets become activated during clumping. The N-terminome and proteolytic profiles of MPS platelets indicates abundant proteolytic activity suggestive perhaps that in fact a range of proteases become activated as the platelets clump. Can the authors indicate which proteases are involved?

2. One patient is aged 81 and her daughter is 65; have they acquired comorbidities associated with the elderly such as diabetes, high blood pressure or arthritis that could influence their current phenotype? What drugs are being taken? At the time of bleeding, what was their whole blood platelet count and platelet volume and was VWF multimer analysis performed? High molecular multimers should be much decreased. It is necessary to provide up-to-date data for these parameters.

3. As the authors correctly state, VWF-mediated platelet clumping is a feature of this disorder. Following from my previous remarks, were platelet clumps present in the blood at the time that PRP was made. I am surprised that the clumps did not sediment with the leukocytes. Were the fluorescence signal intensities shown for the patients in Fig 1A iii (albumin substrate) related to individual platelets or clumps or both? How can they be compared to the controls when clumps were not present? In Fig 1A iv with Fg as substrate the control platelets now expressed more P-selectin. Were the adhered platelets present as single platelets or aggregates? Does the reduced fluorescence for MPS platelets on Fg not indicate a decreased activation of $\alpha\text{IIb}\beta\text{3}$ in the MPS platelets? I think that the following publication should be cited as it concerns the same mutation as the current study (Casari C, Berrou E, Lebret M, Adam F, Kauskot A, Bobe R, Desconclois C, Fressinaud E, Christophe OD, Lenting PJ, Rosa JP, Denis CV, Bryckaert M. von Willebrand factor mutation promotes thrombocytopeny by inhibiting integrin $\alpha\text{IIb}\beta\text{3}$. *J Clin Invest*. 2013

Dec;123(12):5071-81. doi: 10.1172/JCI69458. Epub 2013 Nov 25. PMID: 24270421).

Notwithstanding, the fluorescent images in Fig. 1 are beautiful and the authors are to be congratulated. In Fig. 1D, the results are from proteomic analyses and not fluorescence studies; this is not immediately apparent from the Figure. How these results were calculated is also not readily apparent for platelet suspensions that could contain aggregates, please clarify. Finally do these results suggest that the enlarged MPS platelets have a low content of alpha- and dense granules?

4. Indomethacin was added with apyrase to the PRP prior to the next centrifugation, was this to help dissociate platelet clumps or facilitate resuspension? Some studies were performed on PRP and some on washed platelets. Please clarify this in the Figure legends. It is important to show that the results were representative of all platelets in the blood? Did the authors look at smears taken with whole blood to confirm that clumps were present prior to the centrifugation to prepare PRP? Again, I come back to the question concerning the percentage of the clumps retained in the PRP.

5. One of the early reported characteristics of MPS platelets was a decreased sensitivity to thrombin (see Milton JG, Frojmovic MM, Tang SS; White JG. Spontaneous platelet aggregation in a hereditary giant platelet syndrome (MPS). *Am J Pathol* 1984; 114(2):336-345), a publication that also shows the fine ultrastructural features of the giant platelets. Possibly the patients featured here were also in this report. In the current study, the authors show that the MPS platelets generated less thrombin that is not the same thing and relates to the reduced PS expression. But they unambiguously show that MPS platelets have a reduced response to collagen. In all probability then, the hemostatic activity of MPS platelets is attenuated to all major agonists. In the publication cited above, J.G. White showed that MPD platelets contained giant or fused granules. Were these confirmed in the platelets of the two patients studied here? This may explain the results shown in Fig 1A IV.

6. The results in this manuscript suggest that MPS platelets become exhausted and that increased platelet size and ultrastructural changes linked to the degradation of cytoskeletal proteins are progressively acquired. Indeed major decreases in myosin light chain and tubulin proteins were observed. Yet the timing and the consequences of such decreases remain unclear. Mutations of MYH9 gives rise to macrothrombocytopenia while those affecting tubulin genes have less affect on platelet size but do reduce function. Loss of filamin gives rise to macrothrombocytopenia while dysfunction or loss of talin can render platelets inactive through loss of α IIb β 3 activation. Could the authors please provide a succinct supplementary Table in which they list the degree to which the major contractile proteins of platelets are affected (or not) in MPS. It is impossible to do this at present in view of the mass of data provided in the Suppl Data. Do the authors know if reduced collagen-induced platelet ballooning is affected in MYH9-related disease or is specific to MPS?

7. In a novel part of their work the authors show that CLIC1 (a chloride intracellular channel protein) also has decreased expression in MPS platelets and that reduced chloride influx limits the procoagulant response. This is fine, but in so doing they select one aspect of the platelet procoagulant response without telling us about the others. In the Scott syndrome, an abnormal blood cell procoagulant response is linked to the loss of TMEM6F (ANO6) that mediates phospholipid scrambling but which can also act as a chloride ion channel. Do the authors know if TMEM16F is functioning normally in the MPS platelets? Any anomaly in TMEM16F would also account for decreased PS exposure. Likewise mitochondrial inner membrane desensitization is another major player in phosphatidylserine (PS) exposure on platelets. Is this mechanism also functional in type 2B VWD platelets that indeed show some ultrastructural signs of apoptosis? Did they have a normal mitochondria content? I am sorry to raise this but when I tried to obtain ref 33 on PUBMED I

received a response suggesting that it had been withdrawn. Please check.

8. Have the authors measured the residual prothrombin in the serum of the patients? If there is a defect in procoagulant activity then this should be accompanied by an increase in prothrombin in their serum. Is this the case?

9. In conclusion, two major questions remain for the authors which merit more attention in the discussion. Firstly, when do the changes described in the MPS platelets arise? Are they a consequence of platelet clumping in the circulation or do they arise during megakaryocyte maturation and/or platelet biogenesis? The basic gene defect is VWFV1316M so it is logical to propose that all the platelet abnormalities arise secondarily to the upregulated VWF. Or are there other gene defects to be located in this family (such as in CLIC1). The second major question is whether MPS should continue to be considered as a distinct platelet disorder or whether the defects described here for the Canadian patients will be confirmed worldwide for other families with this mutation or indeed for all families with all VWD2B mutations. I hope that the authors will continue with this work to answer these questions in follow-up reports.

Minor Comment

Reference 5 is incomplete

Reviewer #2 (Remarks to the Author):

Agbani et al describe the characterization of the rare Montreal Platelet Syndrome (MPS) including a proteomic and N-terminomic analysis. They conclude that MPS platelets are “basally activated, partially degranulated, and have marked loss of regulatory, cytoskeletal, and contractile proteins that may account for structural disorganization, giant platelet formation and the reduction in the haemostatic response”.

While studies of rare platelet disorders are generally of interest in order to better understand the fundamentals of platelet (dys)function, the presentation of the results is far from optimal.

Several major concerns need to be addressed in a revision:

- 1) To this reviewer, there seems to be a connection to another proteomics paper where the Scott syndrome has been studied using proteomics, including N-terminomics, in order to better understand the procoagulant response. I wonder why the authors do not mention this study nor the first major platelet proteomics study by Burkhard et al.
- 2) There are some inconsistencies in the methods part. First a lysate preparation right after washed platelet preparation is described, but this is partially in contradiction to the lysis protocol mentioned in the proteomics section. Was this for western blot analyses? Please clarify.
- 3) The proteomics methods section states that DNA was sheared and afterwards pelleted. A major characteristic of platelets is that they are anucleate cell fragments and do not contain DNA. Please clarify.
- 4) The authors write that they topped their SDS containing lysate with GuHCl. In this reviewer’s hand SDS and GuHCl typically precipitate under RT conditions. Please clarify.
- 5) There is literally not a single sentence on (a) how the LC-MS measurements nor (b) the data analysis was performed. The authors discuss about changes in the N-terminome and proteome, but it is impossible to reproduce any of it. It is not sufficient to upload data to a repository, details need to be given at least in a supplemental methods file. This reviewer has never seen this before and

thought they missed it, but 2 files named “proteomics data analysis” and “N-terminomics data analysis” are merely STRING-derived figures that do not even mention the used confidence level nor how candidates were chosen to build the network. This is quite disappointing.

5) The authors used light/heavy formaldehyde labeling, shown in figure 2A as comparison between 1 MPS and 1 control – but there are 2 MPS and 4 controls, please describe how the experimental design was and how the data has been combined and processed. What about missing values? What were criteria to define a “regulated” protein/N-terminus?

6) The supplemental tables with proteomics and N-terminomics data are impossible to read. Normally, authors are supposed to check their files before final submission.

7) Some sentences are hard to understand. What does “These analyses were further validated using STRING-db and enrichment of cell extra-cellular matrix (ECM) interactions, which showed diminished platelet activation and adherens junction in 2B-VWDMPS platelets” mean? How do you validate proteomics data using STRING and how do you validate it using “enrichment of ECM interactions”?

8) “Our N terminomics/TAILS analysis identified 9 different cleaved protein including von Willebrand Factor (VWF) at 1780R↓Y1781, fibronectin (FN1) at 1365R↓F1366 and Crk-like protein (CRKL) at 290H↓V291.” It is unclear how “different cleaved protein” is defined. There are some error bars given for what this reviewer believes are protein expression data from the shotgun analysis (Figure 2E) but some of the error bars seem to be extremely low – please explain and include a more detailed description of the data analysis.

9) In particular figure 1 is very confusing with panels A-E that in case of C are divided into 9 (i-ix) subpanels. The figure legend is extremely long – maybe it would make sense to divide figure 1 into two figures.

10) All proteomics fold-changes are reported as log2. While log2 is certainly useful for data analysis, it would be easier for the reader to report actual fold-changes as not everyone will readily transform log2 into normal fold-changes.

11) The conclusion is extremely short.

Minor issue:

1) The abstract should clearly state that this is an extremely rare disorder right in the first sentence.

UNIVERSITY OF
CALGARY

University of Calgary
Cumming School of Medicine
3330 Hospital Drive NW
Calgary, AB, T2N 4N1

Email: antoine.dufour@ucalgary.ca
Website: www.dufourlab.com

Dear Dr. Andreia Cunha,

We would like to thank the editor and reviewers for handling our manuscript and for the invitation to revise and resubmit it. We are pleased that a reviewer felt that *“The manuscript is very well written and provides a new insight on MPS”*. The reviewers suggested useful changes and additional information to be included in our manuscript. Therefore, we performed a thorough re-write and have addressed all the points raised by the reviewers to the best of our ability. We have re-analyzed our data. We have added additional information throughout the manuscript and how our data was analyzed. We are gratified that these new data and analyses strengthened our previous findings. We thank the reviewers for their excellent insights and constructive comments that have led to a much-improved manuscript.

Reviewer #1:

The authors have studied two members of a family with what was historically termed Montreal platelet syndrome (MPS) and then shown to be von Willebrand disease type 2B (Type 2B VWD) caused by a heterozygous VWFV1316M mutation. Man Chiu Poon has a history of publications on MPS and so knows the disorder well. Combining his experience with the groups of A Dufour (N-terminomics) and A. Poole (procoagulant activity) has enabled a unique look at two members of the original family with this rare platelet disorder and provided some stimulating results. The manuscript is very well written and provides a new insight on MPS. At the same time, the authors have been highly selective with regard to the results presented.

1. Two members of the original family designated as MPS, a mother and her daughter, featured in this study. In view of their age, it would be interesting to know if they were among the original patients studied by Frojmovic and Milton in their original reports on this disease.

Our Response: We have clarified this in the Method section, paragraph 2: “Both were in the original report in 1963 (**reference 8**) and both participated in Frojmovic’s original MPS studies beginning in 1979 (**reference 9**)”.

In fact it would be useful to have in the Supplemental data a list of all previous publications featuring these 2 patients. Enigmatically, the authors fail to cite a highly relevant study performed by Okita et al (Okita JR, Frojmovic MM, Kristopeit S, Wong T, Kunicki TJ. Montreal platelet syndrome: a defect in calcium-activated neutral protease (calpain). Blood 1989; 1:715-721).

This is a great suggestion. We now state the following in **methods section** under sub-heading “The 2B-VWD^{MPS} Patients and Healthy Controls.”

“Eight previous studies have reported on the two patients of the present study^{a-h}. We also provide a list of these studies under references. In addition, we have added these MPS publications to the main texts in chronological order^{a-h} and in supplementary **Table 6**

- a) Lacombe, M. & D, A.G. Etudes sur une thrombopathie familiale. *Nouv Rev Fr Hematol* **3**, 611-614 (1963).

- b) Milton, J.G. & Frojmovic, M.M. Shape-changing agents produce abnormally large platelets in a hereditary "giant platelets syndrome (MPS)". *J Lab Clin Med* **93**, 154-161 (1979).
- c) Frojmovic, M.M. & Milton, J.G. Physical, chemical and functional changes following platelet activation in normal and "giant" platelets. *Blood Cells* **9**, 359-382 (1983).
- d) Milton, J.G., Frojmovic, M.M., Tang, S.S. & White, J.G. Spontaneous platelet aggregation in a hereditary giant platelet syndrome (MPS). *Am J Pathol* **114**, 336-345 (1984).
- e) Milton, J.G., Hutton, R.A., Tuddenham, E.G. & Frojmovic, M.M. Platelet size and shape in hereditary giant platelet syndromes on blood smear and in suspension: evidence for two types of abnormalities. *J Lab Clin Med* **106**, 326-335 (1985).
- f) Okita, J.R., Frojmovic, M.M., Kristopeit, S., Wong, T. & Kunicki, T.J. Montreal platelet syndrome: a defect in calcium-activated neutral proteinase (calpain). *Blood* **74**, 715-721 (1989).
- g) Jackson, S.C., *et al.* The Montreal platelet syndrome kindred has type 2B von Willebrand disease with the VWF V1316M mutation. *Blood* **113**, 3348-3351 (2009).
- h) Poon, M.C., Rand, M.L. & Jackson, S.C. 2B or not to be--the 45-year saga of the Montreal Platelet Syndrome. *Thromb Haemost* **104**, 903-910 (2010)

Were the current patients a part of the Okita study? Either way, this publication must be cited for calpain (a calcium-activated protease) activity may well account for the decreased amount of selected cytoskeletal and other proteins in the platelets of the patients as described in the current report. Was calpain normally detected in the proteomic studies? Do they agree with Okita et al that it is decreased in the platelets of the patients studied? Possibly it becomes used up as the MPS platelets become activated during clumping.

Yes, both mother and daughter in this study participated in Okita's study. The reviewer is correct about Calpain-1 as it was downregulated in the patients' platelets in addition to the Calpain small subunit 1 (CAPNS1) required for Calpain activity. Calpain-1 and CAPNS1 were downregulated in both our shotgun proteomics preTAILS and TAILS data. These data have been added in **NEW Supplementary Table 4**. In discussing our results, we include the statement below:

“Additionally, calpain-1 and the calpain small subunit 1 (CAPNS1) required for calpain activity were downregulated in 2B-VWD^{MPS} platelets as shown by both our shotgun proteomics preTAILS and TAILS data (**supplementary Table 1-3**). This supports the previous report of abnormally low levels of calpain-1 and up to 70% decrease in calpain activity in ‘MPS platelets’¹.”

The N-terminome and proteolytic profiles of MPS platelets indicates abundant proteolytic activity suggestive perhaps that in fact a range of proteases become activated as the platelets clump. Can the authors indicate which proteases are involved?

Great suggestion and yes, we identified 47 proteases and protease inhibitors now included in **NEW Supplementary Table 4**.

2. One patient is aged 81 and her daughter is 65; have they acquired comorbidities associated with the elderly such as diabetes, high blood pressure or arthritis that could influence their current phenotype? What drugs are being taken?

Both 2B-VWD^{MPS} patients were on Atorvastatin, a lipid modifying drug and an anti-angiogenesis agent for GI angiodysplasia. In addition, the mother has hypertension, which was controlled with a calcium channel blocker Amlodipine, and the daughter is non-hypertensive. Importantly, neither of the 2B-VWD^{MPS} patients studied has other comorbidity such as diabetes, renal or rheumatologic problems. We now include this information in the **Method section (paragraph 2)**. Given the low platelet count, we have found it difficult to recover sufficient platelets for aggregation studies. To the best of our knowledge, these two medications did not affect our PMD analysis outcomes, which were comparable between both 2B-VWD^{MPS} patients.

At the time of bleeding, what was their whole blood platelet count and platelet volume and was VWF multimer analysis performed? High molecular multimers should be much decreased. It is necessary to provide up-to-date data for these parameters.

Both patients had persistent platelet clumping irrespective of bleeding status. Platelet counts were for the most part impractical to enumerate; however, when enumerable, counts remained stable during the study period, irrespective of bleeding status (L.T.B: 25 ± 2 , I.B: $29 \pm 3 \times 10^9/L$). We did not repeat multimer studies during the bleeding state as a matter of routine. We now include this information in the Results/ Discussion section. Furthermore, multimer studies are qualitative showing loss of higher multimers, and are not quantitative enough to evaluate clinical changes. Also, coagulation parameters were also stable over time (and overlapping time of our study). See table below for example of recorded values.

	VWF: Activity	VWF: antigen	FVIII
Mother	34	101	85
Daughter	25	110	121

3. As the authors correctly state, VWF-mediated platelet clumping is a feature of this disorder. Following from my previous remarks, were platelet clumps present in the blood at the time that PRP was made. I am surprised that the clumps did not sediment with the leukocytes. Were the fluorescence signal intensities shown for the patients in Fig 1A iii (albumin substrate) related to individual platelets or clumps or both? How can they be compared to the controls when clumps were not present?

This is a valid remark by the reviewer. The image in **Figure 1A-ii** show individual platelets and platelet clumps in PRP of 2B-VWD^{MPS} over serum albumin substrate, indicating the clumps were present at the time that PRP was prepared (Mean \pm SD, HPC: 1.6 ± 1.2 vs 2B-VWD^{MPS}: 20.4 ± 3.7). The fluorescence signal quantified in **Figure 1A-iii** were derived from both individual platelets and platelet clumps as shown in **Figure 1A-ii**. Also, we occasionally sighted heterotypical aggregates of platelets and leukocytes in 2B-VWD^{MPS} but not HPC PRP (data not shown). We now include these comments in the discussion of the results.

In Fig 1A iv with Fg as substrate the control platelets now expressed more P-selectin. Were the adhered platelets present as single platelets or aggregates? Does the reduced fluorescence for MPS platelets on Fg not indicate a decreased activation of $\alpha_{IIb}\beta_3$ in the MPS platelets?

The figure showing data of platelets adherent to fibrinogen (Fg) is now **Figure 1B**. Here, both individual platelets and platelet clumps were analysed, and control platelets expressed more P-selectin with Fg stimulation, indicating a more activated status compared to 2B-VWD^{MPS} platelets. Similarly, the reduced fluorescence for 2B-VWD^{MPS} platelets on Fg is likely indicative of a decreased activation of integrin $\alpha_{IIb}\beta_3$ in the MPS platelets², we however did to assess the degree of activation of these receptors using PAC-1.

I think that the following publication should be cited as it concerns the same mutation as the current study (Casari C, Berrou E, Leuret M, Adam F, Kauskot A, Bobe R, Desconclois C, Fressinaud E, Christophe OD, Lenting PJ, Rosa JP, Denis CV, Bryckaert M. ² J Clin Invest. 2013 Dec;123(12):5071-81. doi: 10.1172/JCI69458. Epub 2013 Nov 25. PMID: 24270421). Notwithstanding, the fluorescent images in Fig. 1 are beautiful and the authors are to be congratulated.

We agree and thank the reviewer for this suggestion. The reduced fluorescence for 2B-VWD^{MPS} platelets on Fg is likely indicative of a decreased activation of integrin $\alpha_{IIb}\beta_3$ in the MPS platelets². We have now cited Casari's paper as **reference 20** (see Results and Discussion section).

In Fig. 1D, the results are from proteomic analyses and not fluorescence studies; this is not immediately apparent from the Figure. How these results were calculated is also not readily apparent for platelet suspensions that could contain aggregates, please clarify.

We thank the reviewer for this observation. We have clarified this mismatch by moving this proteomic data to Figure 2 (now **Figure 2B**). This now focuses Figure 1 entirely on platelet PMD and Fluorescence studies and Figure 2 on proteomics data. The proteomic data in **Figure 2** are based on analysis of washed platelet lysates. Also, we explain how the results were calculated in the Method section under Proteomic Data and Bioinformatic Analysis².

Finally do these results suggest that the enlarged MPS platelets have a low content of alpha- and dense granules?

The reviewer is correct; these results suggest that the enlarged MPS platelets have a low content of specific alpha- and dense granules cargoes such as P-selectin and CD63.

4. Indomethacin was added with apyrase to the PRP prior to the next centrifugation, was this to help dissociate platelet clumps or facilitate resuspension?

Indomethacin is considerably more potent than aspirin as an inhibitor of human platelet prostaglandin synthetase *in vitro*³. Also, while aspirin is essentially an irreversible inhibitor, the inhibition produced by indomethacin is reversible. We used indomethacin as to inhibit platelet activation, secretion, and

aggregation during the high-speed centrifugation process of washed platelet preparation. Likely, indomethacin also facilitates the dissociation of platelet clumps and the resuspension of platelets.

Some studies were performed on PRP and some on washed platelets. Please clarify this in the Figure legends. It is important to show that the results were representative of all platelets in the blood?

All images and results shown are representative of all platelets in the blood of respective participant group. In **Figure 1**, we state that citrated platelet-rich plasma (PRP) from two 2B-VWD^{MPS} patients and four healthy participant controls (HPC) were used. Also in **Figure 2**, we now state that Data (**Figure 2A-F**) are from washed platelet lysates of healthy controls and 2B-VWD^{MPS} patients. In **Figure 2G**, we now indicate that plasma platelets were used. We believe these statements now further clarify which data were derived from PRP or washed platelets.

Did the authors look at smears taken with whole blood to confirm that clumps were present prior to the centrifugation to prepare PRP? Again, I come back to the question concerning the percentage of the clumps retained in the PRP.

We did not look at whole blood smears in this study. Platelet Clumps in whole blood have been reported in these patients and is always present on these two ladies. Several blood counts and smears have been performed with whole blood of these patients over the last 30 years. In all, there was never an instance where platelet clumps were not present and where platelet count or numbers were considered interpretable. The image in Figure 1A-ii show individual platelets and platelet clumps in PRP of 2B-VWD^{MPS} over serum albumin substrate, indicating the clumps were present at the time that PRP was prepared (Mean \pm SD, HPC: 1.6 \pm 1.2 vs 2B-VWD^{MPS}: 20.4 \pm 3.7).

5. One of the early reported characteristics of MPS platelets was a decreased sensitivity to thrombin (see Milton JG, Frojmovic MM, Tang SS; White JG. Spontaneous platelet aggregation in a hereditary giant platelet syndrome (MPS). Am J Pathol 1984; 114(2):336-345), a publication that also shows the fine ultrastructural features of the giant platelets. Possibly the patients featured here were also in this report. In the current study, the authors show that the MPS platelets generated less thrombin that is not the same thing and relates to the reduced PS expression. But they unambiguously show that MPS platelets have a reduced response to collagen. In all probability then, the hemostatic activity of MPS platelets is attenuated to all major agonists. In the publication cited above, J.G. White showed that MPD platelets contained giant or fused granules. Were these confirmed in the platelets of the two patients studied here? This may explain the results shown in Fig 1A IV.

We thank the author for this interesting remark. Our study showed decreased thrombin generation. However, we have not studied the sensitivity of the 2B-VWD^{MPS} platelets to thrombin. And we did not measure granule size in our study patients.

6. The results in this manuscript suggest that MPS platelets become exhausted and that increased platelet size and ultrastructural changes linked to the degradation of cytoskeletal proteins are progressively acquired. Indeed major decreases in myosin light chain and tubulin proteins were observed. Yet the timing and the consequences of such decreases remain unclear. Mutations of MYH9 gives rise to macrothrombocytopenia while those affecting tubulin genes have less effect on

University of Calgary
Cumming School of Medicine
3330 Hospital Drive NW
Calgary, AB, T2N 4N1

Email: antoine.dufour@ucalgary.ca
Website: www.dufourlab.com

platelet size but do reduce function. Loss of filamin gives rise to macrothrombocytopenia while dysfunction or loss of talin can render platelets inactive through loss of α IIb β 3 activation. Could the authors please provide a succinct supplementary Table in which they list the degree to which the major contractile proteins of platelets are affected (or not) in MPS. It is impossible to do this at present in view of the mass of data provided in Suppl Data.

This is a great point. We have now added these data in **Supplementary Table 5**. We have identified 21 contractile proteins elevated in healthy platelets. These proteins are therefore reduced in the VWD patient's platelets. We identified 8 other contractile proteins elevated in the VWD patient's platelets. Therefore, our data suggest that the VWD patient's platelets are defective in major contractile proteins.

Do the authors know if reduced collagen-induced platelet ballooning is affected in MYH9-related disease or is specific to MPS?

In this study we have demonstrated reductions in collagen induced platelet ballooning in 2B-VWDMPS patients. We do not know if this will also be the case in MYH9 related diseases. However, we opine that it is unlikely, and for the reason discussed below.

The platelet cytoskeleton consists of microtubules, actin and intermediate filaments which give platelet its shape, and their remodelling during activation is necessary for shape change⁴. Several actin-binding proteins including filamin A, α -actinin, gelsolin and tropomyosin, also contribute to the cytoskeleton⁴. Remodeling of the actomyosin-complex is pivotal to clot retraction and is lost or reduced in patients and animal models with mutations in *MYH9*^{5,6}. Conceivably, the *MYH9* gene may play a role in the translocation of procoagulant platelets by pro-aggregatory and contractile platelet phenotype during thrombus formation⁷. During platelet ballooning, the membrane-cytoskeleton interaction is weakened, and the combination of internal hydrostatic and external osmotic pressures irreversibly drive the protrusions of the delaminated platelet membrane^{8,9}. At present, no study has examined the *MYH9* gene role in platelet membrane ballooning; however, since ballooning require a disruption or unwinding of the platelet microtubules system^{8,10}, mutations in *MYH9* is unlikely to impede ballooning.

7. In a novel part of their work the authors show that CLIC1 (a chloride intracellular channel protein) also has decreased expression in MPS platelets and that reduced chloride influx limits the procoagulant response. This is fine, but in so doing they select one aspect of the platelet procoagulant response without telling us about the others. In the Scott syndrome, an abnormal blood cell procoagulant response is linked to the loss of TMEM16F (ANO6) that mediates phospholipid scrambling but which can also acts as a chloride ion channel. Do the authors know if TMEM16F is functioning normally in the MPS platelets? Any anomaly in TMEM16F would also account for decreased PS exposure.

We did not identify this protein in our proteomics data. It does not mean that it is not functioning normally or not, it only does not provide this information in our data. We can subsequently further explore the role of TMEM16F in our next study.

University of Calgary
Cumming School of Medicine
3330 Hospital Drive NW
Calgary, AB, T2N 4N1

Email: antoine.dufour@ucalgary.ca

Website: www.dufourlab.com

Likewise mitochondrial inner membrane desensitization is another major player in phosphatidylserine (PS) exposure on platelets. Is this mechanism also functional in type 2B VWD platelets that indeed show some ultrastructural signs of apoptosis? Did they have a normal mitochondria content? I am sorry to raise this but when I tried to obtain ref 33 on PUBMED I received a response suggesting that it had been withdrawn. Please check.

Platelet PS exposure can be facilitated by the agonist-mediated mitochondrial permeability transition pore (mPTP) opening and cytoplasmic Ca^{2+} dependent activation of the scramblase TMEM16f (Anoctamin 6)¹¹⁻¹³. Cyclophilin D is involved in formation of the mPTP, and its inhibition reduces pore formation/opening, necrosis¹⁴, and the formation of procoagulant platelets¹⁵. In this study, we did not determine whether mPTP was dysfunctional or to what extent CypD dependent mPTP opening contribute to procoagulant potentials in 2B-VWD platelets. This will be a subject for future investigation.

Also, **reference 33 (now 38)** cited below, can be retrieved from PUBMED using the following: PMID: 20049953 DOI: 10.1002/dvg.20590.

Generation and characterization of mice with null mutation of the chloride intracellular channel 1 gene. Min Ru Qiu, Lele Jiang, Klaus I Matthaei, Simone M Schoenwaelder, Tamara Kuffner, Pierre Mangin, Joanne E Joseph, Joyce Low, David Connor, Stella M Valenzuela, Paul M G Curmi, Louise J Brown, Martyn Mahaut-Smith, Shaun P Jackson, Samuel N Breit. *Genesis* 2010 **48(2):127-36.**

8. Have the authors measured the residual prothrombin in the serum of the patients?

No, we did not measure the residual prothrombin the serum of the patients. Indeed, we do not have any serum stored and it is now difficult to obtain additional blood samples from these patients.

If there is a defect in procoagulant activity then this should be accompanied by an increase in prothrombin in their serum. Is this the case?

Our proteomic analysis showed that compared to controls, platelet α -granules of 2B-VWD^{MPS} patients contained nearly twice as much prothrombin the inactive precursor of thrombin¹⁶, still plasma levels showed a fold decrease in prothrombin, and prothrombin activation peptide fragment 1+2, thrombin light chain and thrombin heavy chain. This may be indicative of a dysfunction in granular secretion and is consistent with defects in procoagulant activity observed in 2B-VWD^{MPS} platelets. We now include these comments in the discussion of our results.

9. In conclusion, two major questions remain for the authors which merit more attention in the discussion. Firstly, when do the changes described in the MPS platelets arise? Are they a consequence of platelet clumping in the circulation, or do they arise during megakaryocyte maturation and/or platelet biogenesis?

We now include the following statement in the **discussion section**.

University of Calgary
Cumming School of Medicine
3330 Hospital Drive NW
Calgary, AB, T2N 4N1

Email: antoine.dufour@ucalgary.ca

Website: www.dufourlab.com

“Overall, our data does not suggest whether the changes in the MPS platelets we described in this report occurred because of the ‘clumping process’ of circulating platelets or whether they arise during megakaryocyte maturation and/or platelet biogenesis. We however speculate that these changes are more likely the consequence of the latter, and Nurdens et al have reported data showing defective megakaryopoiesis in patients with 2B-VWD^{17,18}. Studies are now needed to clarify this.”

The basic gene defect is VWFV1316M so it is logical to propose that all the platelet abnormalities arise secondarily to the upregulated VWF. Or are there other gene defects to be located in this family (such as in CLIC1).

The basic gene defect is VWFV1316M; In addition, our study now indicates a reduced expression of the chloride channel CLIC1. New studies will be required to demonstrate whether MPS patients have additional platelet related gene mutations.

The second major question is whether MPS should continue to be considered as a distinct platelet disorder or whether the defects described here for the Canadian patients will be confirmed worldwide for other families with this mutation or indeed for all families with all VWD2B mutations. I hope that the authors will continue with this work to answer these questions in follow-up reports.

We thank the reviewer for this interesting remark. We hope our study represents the initial efforts to address these questions.

Minor Comment

Reference 5 is incomplete

Thank you, we have updated reference 5.

References

1. Okita, J.R., Frojmovic, M.M., Kristopeit, S., Wong, T. & Kunicki, T.J. Montreal platelet syndrome: a defect in calcium-activated neutral proteinase (calpain). *Blood* **74**, 715-721 (1989).
2. Casari, C., *et al.* von Willebrand factor mutation promotes thrombocytopeny by inhibiting integrin α IIb β 3. *J Clin Invest* **123**, 5071-5081 (2013).
3. Crook, D. & Collins, A.J. Comparison of effects of aspirin and indomethacin on human platelet prostaglandin synthetase. *Annals of the Rheumatic Diseases* **36**, 459-463 (1977).
4. Nurden, A.T. Molecular basis of clot retraction and its role in wound healing. *Thrombosis Research* (2022).
5. Baumann, J., *et al.* Reduced platelet forces underlie impaired hemostasis in mouse models of MYH9-related disease. *Sci Adv* **8**, eabn2627 (2022).
6. Léon, C., *et al.* Megakaryocyte-restricted MYH9 inactivation dramatically affects hemostasis while preserving platelet aggregation and secretion. *Blood* **110**, 3183-3191 (2007).
7. Nechipurenko Dmitry, Y., *et al.* Clot Contraction Drives the Translocation of Procoagulant Platelets to Thrombus Surface. *Arteriosclerosis, Thrombosis, and Vascular Biology* **39**, 37-47 (2019).
8. Agbani, E.O. & Poole, A.W. Procoagulant Platelets:-Generation, Function and Therapeutic Targeting in Thrombosis. *Blood* **130**, 2171-2179 (2017).
9. Charras, G. & Paluch, E. Blebs lead the way: how to migrate without lamellipodia. *Nat Rev Mol Cell Biol* **9**, 730-736 (2008).
10. Agbani, E.O., *et al.* Coordinated Membrane Ballooning and Procoagulant Spreading in Human Platelets. *Circulation* **132**, 1414-1424 (2015).
11. Fernández, D.I., Kuijpers, M.J.E. & Heemskerk, J.W.M. Platelet calcium signaling by G-protein coupled and ITAM-linked receptors regulating anoctamin-6 and procoagulant activity. *Platelets* **32**, 863-871 (2021).
12. Grubb, S.r., *et al.* TMEM16F (Anoctamin 6), an anion channel of delayed Ca²⁺ activation. *The Journal of General Physiology* **141**, 585-600 (2013).
13. Jobe, S.M., *et al.* Critical role for the mitochondrial permeability transition pore and cyclophilin D in platelet activation and thrombosis. *Blood* **111**, 1257 (2008).
14. Nakagawa, T., *et al.* Cyclophilin D-dependent mitochondrial permeability transition regulates some necrotic but not apoptotic cell death. *Nature* **434**, 652-658 (2005).
15. Schoenwaelder, S.M., *et al.* Two distinct pathways regulate platelet phosphatidylserine exposure and procoagulant function. *Blood* **114**, 663-666 (2009).
16. Blair, P. & Flaumenhaft, R. Platelet alpha-granules: basic biology and clinical correlates. *Blood Rev* **23**, 177-189 (2009).
17. Nurden, P., *et al.* Abnormal VWF modifies megakaryocytopoiesis: studies of platelets and megakaryocyte cultures from patients with von Willebrand disease type 2B. *Blood* **115**, 2649-2656 (2010).
18. Nurden, P., *et al.* Platelet morphological changes in 2 patients with von Willebrand disease type 3 caused by large homozygous deletions of the von Willebrand factor gene. *Haematologica* **94**, 1627-1629 (2009).

19. Burkhardt, J.M., *et al.* The first comprehensive and quantitative analysis of human platelet protein composition allows the comparative analysis of structural and functional pathways. *Blood* **120**, e73-e82 (2012).
20. Solari, F.A., *et al.* Combined Quantification of the Global Proteome, Phosphoproteome, and Proteolytic Cleavage to Characterize Altered Platelet Functions in the Human Scott Syndrome. *Molecular & Cellular Proteomics* **15**, 3154-3169 (2016).
21. Szklarczyk, D., *et al.* The STRING database in 2021: customizable protein-protein networks, and functional characterization of user-uploaded gene/measurement sets. *Nucleic Acids Res* **49**, D605-d612 (2021).

Reviewer #2:

Agbani et al describe the characterization of the rare Montreal Platelet Syndrome (MPS) including a proteomic and N-terminomic analysis. They conclude that MPS platelets are “basally activated, partially degranulated, and have marked loss of regulatory, cytoskeletal, and contractile proteins that may account for structural disorganization, giant platelet formation and the reduction in the haemostatic response”. While studies of rare platelet disorders are generally of interest in order to better understand the fundamentals of platelet (dys)function, the presentation of the results is far from optimal.

Several major concerns need to be addressed in a revision:

1) To this reviewer, there seems to be a connection to another proteomics paper where the Scott syndrome has been studied using proteomics, including N-terminomics, in order to better understand the procoagulant response. I wonder why the authors do not mention this study nor the first major platelet proteomics study by Burkhardt et al.

Great suggestion and it was an oversight on our part. We have included reference to these papers^{19,20} in the methods and discussion sections of the paper.

2) There are some inconsistencies in the methods part. First a lysate preparation right after washed platelet preparation is described, but this is partially in contradiction to the lysis protocol mentioned in the proteomics section. Was this for western blot analyses? Please clarify.

The method section has been updated and it is now clarified. Sorry for the confusion. The section “Washed platelets and Lysate Preparation for Protein Assays.” has been modified and it was for Western blot analysis and other protein assays.

3) The proteomics methods section states that DNA was sheared and afterwards pelleted. A major characteristic of platelets is that they are anucleate cell fragments and do not contain DNA. Please clarify.

Yes, the reviewer is correct about the DNA in platelets. We typically always sonicate our samples as sometimes, there are condensate during the lysis step and sonicate often helps to clarify our lysis

solution or reduce some minor clumps that can occur. We have updated our method section and remove the shearing DNA from our platelets.

4) The authors write that they topped their SDS containing lysate with GuHCl. In this reviewer's hand SDS and GuHCl typically precipitate under RT conditions. Please clarify.

Yes, this was an oversight and no GuHCL was added and only SDS containing lysate buffer. The Method section has been updated.

5) There is literally not a single sentence on (a) how the LC-MS measurements nor (b) the data analysis was performed. The authors discuss about changes in the N-terminome and proteome, but it is impossible to reproduce any of it. It is not sufficient to upload data to a repository, details need to be given at least in a supplemental methods file. This reviewer has never seen this before and thought they missed it, but 2 files named "proteomics data analysis" and "N-terminomics data analysis" are merely STRING-derived figures that do not even mention the used confidence level nor how candidates were chosen to build the network. This is quite disappointing.

We are sorry for the oversight as we had a word limit for our initial submission. **We have added all this information in the method section** (and it can be moved as supplementary methods if it is too long to be included in the original manuscript). You can find the information for the LC-MS/MS analysis, proteomic data and bioinformatics analysis, Reactome and STRING-db analysis, and the statistical analysis.

5) The authors used light/heavy formaldehyde labeling, shown in figure 2A as comparison between 1 MPS and 1 control – but there are 2 MPS and 4 controls, please describe how the experimental design was and how the data has been combined and processed.

We ran the two MPS patient samples against the first two consecutive healthy controls of the study. We had enough platelets to set up another MPS run against two other healthy controls. Therefore, we had data from 2 patients against 2 controls, and we were able to repeat the 2 patients samples against 2 new controls.

What about missing values? What were criteria to define a "regulated" protein/N-terminus?

We set the cut-off as 2 identified peptides out of 3.

6) The supplemental tables with proteomics and N-terminomics data are impossible to read. Normally, authors are supposed to check their files before final submission.

We are sorry for the output issues. We had prepared all our TAILS and proteomics data in an excel spreadsheet to make it easy to read. Is it possible that it was converted (to PDF) during the submission process and sent as a PDF which sometimes occur? We will ask the editor to send out the excel

spreadsheets for all supplementary tables to make it readable for the revision. All supplementary tables in excel format have been uploaded and should be easy to read.

7) Some sentences are hard to understand. What does “These analyses were further validated using STRING-db and enrichment of cell extra-cellular matrix (ECM) interactions, which showed diminished platelet activation and adherens junction in 2B-VWDMPS platelets” mean?

Sorry for the confusion, we have now rewritten and clarify this sentence. It now states: “We performed additional pathway enrichment analyses using another tool, STRING-db²¹, and we also identified an enrichment of cell-extra-cellular matrix (ECM) interactions. These data suggests that diminished platelet activation and adherens junction were detected in 2B-VWD^{MPS} platelets (**Supplementary Figure 1B-C**)”

How do you validate proteomics data using STRING and how do you validate it using “enrichment of ECM interactions”?

We have now clarified this sentence. We agree with the reviewer, and we can validate proteomics data this way and we now state that identified an enrichment of ECM interactions.

8) “Our N terminomics/TAILS analysis identified 9 different cleaved protein including von Willebrand Factor (VWF) at 1780R↓Y1781, fibronectin (FN1) at 1365R↓F1366 and Crk-like protein (CRKL) at 290H↓V291.” It is unclear how “different cleaved protein” is defined.

We described cleaved protein as a neo-N-termini detected in the TAILS data (Supplementary Table 3) that is changed based on our boxplot interquartile analysis. The sentence has been clarified in the result section.

There are some error bars given for what this reviewer believes are protein expression data from the shotgun analysis (Figure 2E) but some of the error bars seem to be extremely low – please explain and include a more detailed description of the data analysis.

This is a valid point. Our proteomics experiments were conducted in replicates for each patient and control combination. The error bars in **Figure 2E** (now **2F**) represent the standard error of means of the data. We now revised this to SD and clarify in the Figure legend and Methods section. Some of the error bars seem to be extremely low because values in replicate experiments were very similar.

9) In particular figure 1 is very confusing with panels A-E that in case of C are divided into 9 (i-ix) subpanels. The figure legend is extremely long – maybe it would make sense to divide figure 1 into two figures.

Unfortunately, were limited by the Journal’s requirement for the number of Figures in this report. Request for additional data and clarification during review process have also limited our ability to reduce the number of Figures or length of Figure Legend. To enhance clarity, we have now

University of Calgary
Cumming School of Medicine
3330 Hospital Drive NW
Calgary, AB, T2N 4N1

Email: antoine.dufour@ucalgary.ca

Website: www.dufourlab.com

restructured both Figure 1 and 2 slightly. For example, we moved **Figure 1D** (now **Figure 2B**) to Figure 2. We have thus been able to focus Figure 1 and 2 on mainly platelet PMD and proteomics, respectively.

10) All proteomics fold-changes are reported as log2. While log2 is certainly useful for data analysis, it would be easier for the reader to report actual fold-changes as not everyone will readily transform log2 into normal fold-changes.

We agree with this remark, and we have added the fold changes in the supplementary tables.

11) The conclusion is extremely short.

The study conclusion is now extended and revised in accordance with the journal requirements.

Minor issue:

1) The abstract should clearly state that this is an extremely rare disorder right in the first sentence.

Agreed and it was added to the abstract.

REVIEWERS' COMMENTS:

Reviewer #1 (Remarks to the Author):

No further scientific comments but the manuscript as sent is a mess with some supplemental tables no longer present or not mentioned in the text. The Supplemental data begins with Suppl Table 6 when normally this should begin with Suppl Table 1. This is all a shame as the manuscript is of a high level and is of much scientific interest.

Reviewer #2 (Remarks to the Author):

The authors have addressed all my concerns accordingly and I recommend acceptance of the manuscript. Thank you.

University of Calgary
Cumming School of Medicine
3330 Hospital Drive NW
Calgary, AB, T2N 4N1
Email: antoine.dufour@ucalgary.ca
Website: www.dufourlab.com

Dear Dr. Andreia Cunha,

We would like to thank the editor and reviewers for handling our manuscript and for the invitation to send the final version. We are pleased that the reviewers accepted our revision. We thank the reviewers for their excellent insights and constructive comments that have led to a much-improved manuscript.

REVIEWERS' COMMENTS:

Reviewer #1 (Remarks to the Author):

No further scientific comments but the manuscript as sent is a mess with some supplemental tables no longer present or not mentioned in the text. The Supplemental data begins with Suppl Table 6 when normally this should begin with Suppl Table 1. This is all a shame as the manuscript is of a high level and is of much scientific interest.

We have clarified this and have cited and updated all supplementary tables for the manuscript.

Reviewer #2 (Remarks to the Author):

The authors have addressed all my concerns accordingly and I recommend acceptance of the manuscript. Thank you.

We thank the reviewer for their suggestions.